# Inter- and intra-animal variation in the integrative properties of stellate cells in the medial entorhinal cortex

**Hugh Pastoll[1], Derek L Garden[1], Ioannis Papastathopoulos[2,3], Gülşen Sürmeli[1], Matthew F Nolan[1]\***

[1]Centre for Discovery Brain Sciences, University of Edinburgh, Edinburgh, United Kingdom; [2]The Alan Turing Institute, London, United States; [3]School of Mathematics, Maxwell Institute and Centre for Statistics, University of Edinburgh, Edinburgh, United Kingdom

**Abstract** Distinctions between cell types underpin organizational principles for nervous system function. Functional variation also exists between neurons of the same type. This is exemplified by correspondence between grid cell spatial scales and the synaptic integrative properties of stellate cells (SCs) in the medial entorhinal cortex. However, we know little about how functional variability is structured either within or between individuals. Using ex-vivo patch-clamp recordings from up to 55 SCs per mouse, we found that integrative properties vary between mice and, in contrast to the modularity of grid cell spatial scales, have a continuous dorsoventral organization. Our results constrain mechanisms for modular grid firing and provide evidence for inter-animal phenotypic variability among neurons of the same type. We suggest that neuron type properties are tuned to circuit-level set points that vary within and between animals.

## Introduction

The concept of cell types provides a general organizing principle for understanding biological structures including the brain (*Regev et al., 2017*; *Zeng and Sanes, 2017*). The simplest conceptualization of a neuronal cell type, as a population of phenotypically similar neurons with features that cluster around a single set point (*Wang et al., 2011b*), is extended by observations of variability in cell type features, suggesting that some neuronal cell types may be conceived as clustering along a line rather than around a point in a feature space (*Cembrowski and Menon, 2018*; *O'Donnell and Nolan, 2011*; *Figure 1A*). Correlations between the functional organization of sensory, motor and cognitive circuits and the electrophysiological properties of individual neuronal cell types suggest that this feature variability underlies key neural computations (*Adamson et al., 2002*; *Angelo et al., 2012*; *Fletcher and Williams, 2019*; *Garden et al., 2008*; *Giocomo et al., 2007*; *Kuba et al., 2005*; *O'Donnell and Nolan, 2011*). However, within-cell type variability has typically been deduced by combining data obtained from multiple animals. By contrast, the structure of variation within individual animals or between different animals has received little attention. For example, apparent clustering of properties along lines in feature space could reflect a continuum of set points, or could result from a small number of discrete set points that are obscured by inter-animal variation (*Figure 1B*). Moreover, although investigations of invertebrate nervous systems show that set points may differ between animals (*Goaillard et al., 2009*), it is not clear whether mammalian neurons exhibit similar phenotypic diversity (*Figure 1B*). Distinguishing these possibilities requires many more electrophysiological observations for each animal than are obtained in typical studies.

Stellate cells in layer 2 (SCs) of the medial entorhinal cortex (MEC) provide a striking example of correspondence between functional organization of neural circuits and variability of

**\*For correspondence:**
mattnolan@ed.ac.uk

**Competing interests:** The authors declare that no competing interests exist.

**eLife digest** The brain consists of many types of cells that are specialised to perform different tasks. This is similar to how different groups of people will have different responsibilities in a large company. But within each group with the same role, individual employees will also do their jobs in different ways. Does the same apply to the brain? In other words, do individual neurons of the same type – with the same role – process information differently?

To find out, Pastoll et al. studied stellate cells in the mouse brain: these neurons take their name from their distinctive star-shaped arrays of projections, and they work together in groups known as modules to help animals navigate their environment. To determine whether stellate cells differ between mice, and how they might differ within a single animal, Pastoll et al. measured the activity of more than 800 stellate cells in more than two dozen individuals.

The results revealed that stellate cells process the same information differently between mice, which may contribute to variations in behaviour across the species. But even within an individual, stellate cells also showed differences in information processing. In fact, the properties of the stellate cells within each mouse varied along a continuum. This discovery rules out several previous theories on how stellate cells form the modules that support navigation.

The work by Pastoll et al. helps to understand how the brain supports thinking and memory. In the long term, these findings could also have implications for treating brain disorders, as they suggest that variations between people in the properties of their neurons could lead to variations in drug response. Researchers may need to take inter-individual differences into account when planning experiments, and ultimately when designing drugs.

electrophysiological features within a single cell type. The MEC contains neurons that encode an animal's location through grid-like firing fields (*Fyhn et al., 2004*). The spatial scale of grid fields follows a dorsoventral organization (*Hafting et al., 2005*), which is mirrored by a dorsoventral organization in key electrophysiological features of SCs (*Boehlen et al., 2010*; *Dodson et al., 2011*; *Garden et al., 2008*; *Giocomo et al., 2007*; *Giocomo and Hasselmo, 2008a*; *Pastoll et al., 2012a*). Grid cells are further organized into discrete modules (*Stensola et al., 2012*), with the cells within a module having a similar grid scale and orientation (*Barry et al., 2007*; *Gu et al., 2018*; *Stensola et al., 2012*; *Yoon et al., 2013*); progressively more ventral modules are composed of cells with wider grid spacing (*Stensola et al., 2012*). Studies that demonstrate dorsoventral organization of integrative properties of SCs have so far relied on the pooling of relatively few measurements per animal. Hence, it is unclear whether the organization of these cellular properties is modular, as one might expect if they directly set the scale of grid firing fields in individual grid cells (*Giocomo et al., 2007*). The possibility that set points for electrophysiological properties of SCs differ between animals has also not been considered previously.

Evaluation of variability between and within animals requires statistical approaches that are not typically used in single-cell electrophysiological investigations. Given appropriate assumptions, inter-animal differences can be assessed using mixed effect models that are well established in other fields (*Baayen et al., 2008*; *Geiler-Samerotte et al., 2013*). Because tests of whether data arise from modular as opposed to continuous distributions have received less general attention, to facilitate detection of modularity using relatively few observations, we introduce a modification of the gap statistic algorithm (*Tibshirani et al., 2001*) that estimates the number of modes in a dataset while controlling for observations expected by chance (see 'Materials and methods' and *Figure 1— figure supplements 1–5*). This algorithm performs well compared with discreteness metrics that are based on the standard deviation of binned data (*Giocomo et al., 2014*; *Stensola et al., 2012*), which we find are prone to high false-positive rates (*Figure 1—figure supplement 4A*). We find that recordings from approximately 30 SCs per animal should be sufficient to detect modularity using the modified gap statistic algorithm and given the experimentally observed separation between grid modules (see 'Materials and methods' and *Figure 1—figure supplements 2– 3*). Although methods for high-quality recording from SCs in ex-vivo brain slices are well established (*Pastoll et al., 2012b*), typically fewer than five recordings per animal were made in previous studies,

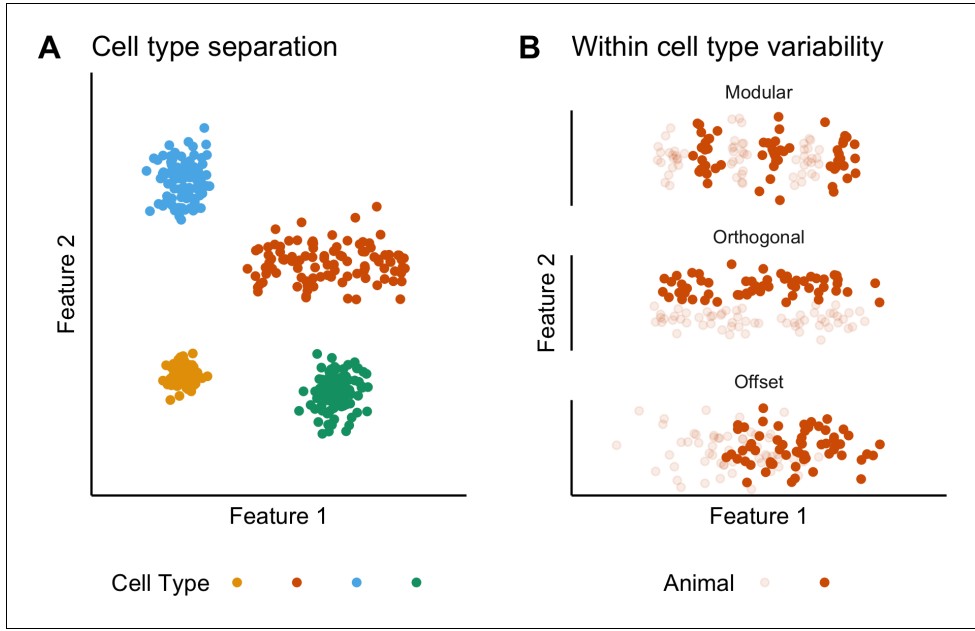

**Figure 1.** Classification and variability of neuronal cell types. (**A**) Neuronal cell types are identifiable by features clustering around a distinct point (blue, green and yellow) or a line (orange) in feature space. The clustering implies that neuron types are defined by either a single set point (blue, green and yellow) or by multiple set points spread along a line (orange). (**B**) Phenotypic variability of a single neuron type could result from distinct set points that subdivide the neuron type but appear continuous when data from multiple animals are combined (modular), from differences in components of a set point that do not extend along a continuum but that in different animals cluster at different locations in feature space (orthogonal), or from differences between animals in the range covered by a continuum of set points (offset). These distinct forms of variability can only be made apparent by measuring the features of many neurons from multiple animals.

The online version of this article includes the following figure supplement(s) for figure 1:

**Figure supplement 1.** A quantitative adaptation of the gap statistic clustering algorithm.

**Figure supplement 2.** Discrimination of continuous from modular organizations using the adapted gap statistic algorithm.

**Figure supplement 3.** Additional evaluation of the adapted gap statistic algorithm.

**Figure supplement 4.** Comparing the adapted gap statistic algorithm with discontinuity measures for discreteness.

**Figure supplement 5.** Evaluation of modularity of grid firing using an adapted gap statistic algorithm.

which is many fewer than our estimate of the minimum number of observations required to test for modularity.

We set out to establish the nature of the set points that establish the integrative properties of SCs by measuring intra- and inter-animal variation in key electrophysiological features using experiments that maximize the number of SCs recorded per animal. Our results suggest that set points for individual features of a neuronal cell type are established at the level of neuronal cell populations, differ between animals and follow a continuous organization.

## Results

### Sampling integrative properties from many neurons per animal

Before addressing intra- and inter-animal variability, we first describe the data set used for the analyses that follow. We established procedures to facilitate the recording of integrative properties of many SCs from a single animal (see 'Materials and methods'). With these procedures, we measured and analyzed electrophysiological features of 836 SCs (n/mouse: range 11–55; median = 35) from 27 mice (median age = 37 days, age range = 18–57 days). The mice were housed either in a standard home cage (dimensions: 0.2 × 0.37 m, N = 18 mice, n = 583 neurons) or from postnatal day 16 in a

2.4 × 1.2 m cage, which provided a large environment that could be freely explored (N = 9, n = 253, median age = 38 days) (*Figure 2—figure supplement 1*). For each neuron, we measured six sub-threshold integrative properties (*Figure 2A–B*) and six supra-threshold integrative properties (*Figure 2C*). Unless indicated otherwise, we report the analysis of datasets that combine the groups of mice housed in standard and large home cages and that span the full range of ages.

Because SCs are found intermingled with pyramidal cells in layer 2 (L2PCs), and as misclassification of L2PCs as SCs would probably confound investigation of intra-SC variation, we validated our criteria for distinguishing each cell type. To establish characteristic electrophysiological properties of L2PCs, we recorded from neurons in layer 2 that were identified by Cre-dependent marker expression in a *Wfs1*Cre mouse line (*Sürmeli et al., 2015*). Expression of Cre in this line, and in a similar line (*Kitamura et al., 2014*), labels L2PCs that project to the CA1 region of the hippocampus, but does not label SCs (*Kitamura et al., 2014*; *Sürmeli et al., 2015*). We identified two populations of neurons in layer 2 of MEC that were labelled in *Wfs1*Cre mice (*Figure 3A–C*). The more numerous population had properties consistent with L2PCs (*Figure 3A,G*) and could be separated from the unidentified population on the basis of a lower rheobase (*Figure 3C*). The unidentified population had firing properties that were typical of layer 2 interneurons (*Gonzalez-Sulser et al., 2014*). A principal component analysis (PCA) (*Figure 3D–F*) clearly separated the L2PC population from the SC population, but did not identify subpopulations of SCs. The properties of the less numerous population were also clearly distinct from those of SCs (*Figure 3A,C*). These data demonstrate that the SC population used for our analyses is distinct from other cell types also found in layer 2 of the MEC.

To further validate the large SC dataset, we assessed the location-dependence of individual electrophysiological features, several of which have previously been found to depend on the dorsoventral location of the recorded neuron (*Boehlen et al., 2010*; *Booth et al., 2016*; *Garden et al., 2008*; *Giocomo et al., 2007*; *Pastoll et al., 2012a*; *Yoshida et al., 2013*). We initially fit the dependence of each feature on dorsoventral position using a standard linear regression model. We found substantial (adjusted $R^2$ >0.1) dorsoventral gradients in input resistance, sag, membrane time constant, resonant frequency, rheobase and the current-frequency (I-F) relationship (*Figure 3G*). In contrast to the situation in SCs, we did not find evidence for dorsoventral organization of these features in L2PCs (*Figure 3G*). Thus, our large dataset replicates the previously observed dependence of integrative properties of SCs on their dorsoventral position, and shows that this location dependence further distinguishes SCs from L2PCs.

## Inter-animal differences in the intrinsic properties of stellate cells

To what extent does variability between the integrative properties of SCs at a given dorsoventral location arise from differences between animals? Comparing specific features between individual animals suggested that their distributions could be almost completely non-overlapping, despite consistent and strong dorsoventral tuning (*Figure 4A*). If this apparent inter-animal variability results from the random sampling of a distribution determined by a common underlying set point, then fitting the complete data set with a mixed model in which animal identity is included as a random effect should reconcile the apparent differences between animals (*Figure 4B*). In this scenario, the conditional $R^2$ estimated from the mixed model, in other words, the estimate of variance explained by animal identity and location, should be similar to the marginal $R^2$ value, which indicates the variance explained by location only. By contrast, if differences between animals contribute to experimental variability, the mixed model should predict different fitting parameters for each animal, and the estimated conditional $R^2$ should be greater than the corresponding marginal $R^2$ (*Figure 4C*).

Fitting the experimental measures for each feature with mixed models suggests that differences between animals contribute substantially to the variability in properties of SCs. In contrast to simulated data in which inter-animal differences are absent (*Figure 4B*), differences in fits between animals remained after fitting with the mixed model (*Figure 4D*). This corresponds with expectations from fits to simulated data containing inter-animal variability (*Figure 4C*). To visualize inter-animal variability for all measured features, we plot for each animal the intercept of the model fit (I), the predicted value at a location 1 mm ventral from the intercept (I+S), and the slope (lines) (*Figure 4E*). Strikingly, even for features such as rheobase and input resistance (IR) that are highly tuned to a neurons' dorsoventral position, the extent of variability between animals is similar to the extent to which the property changes between dorsal and mid-levels of the MEC.

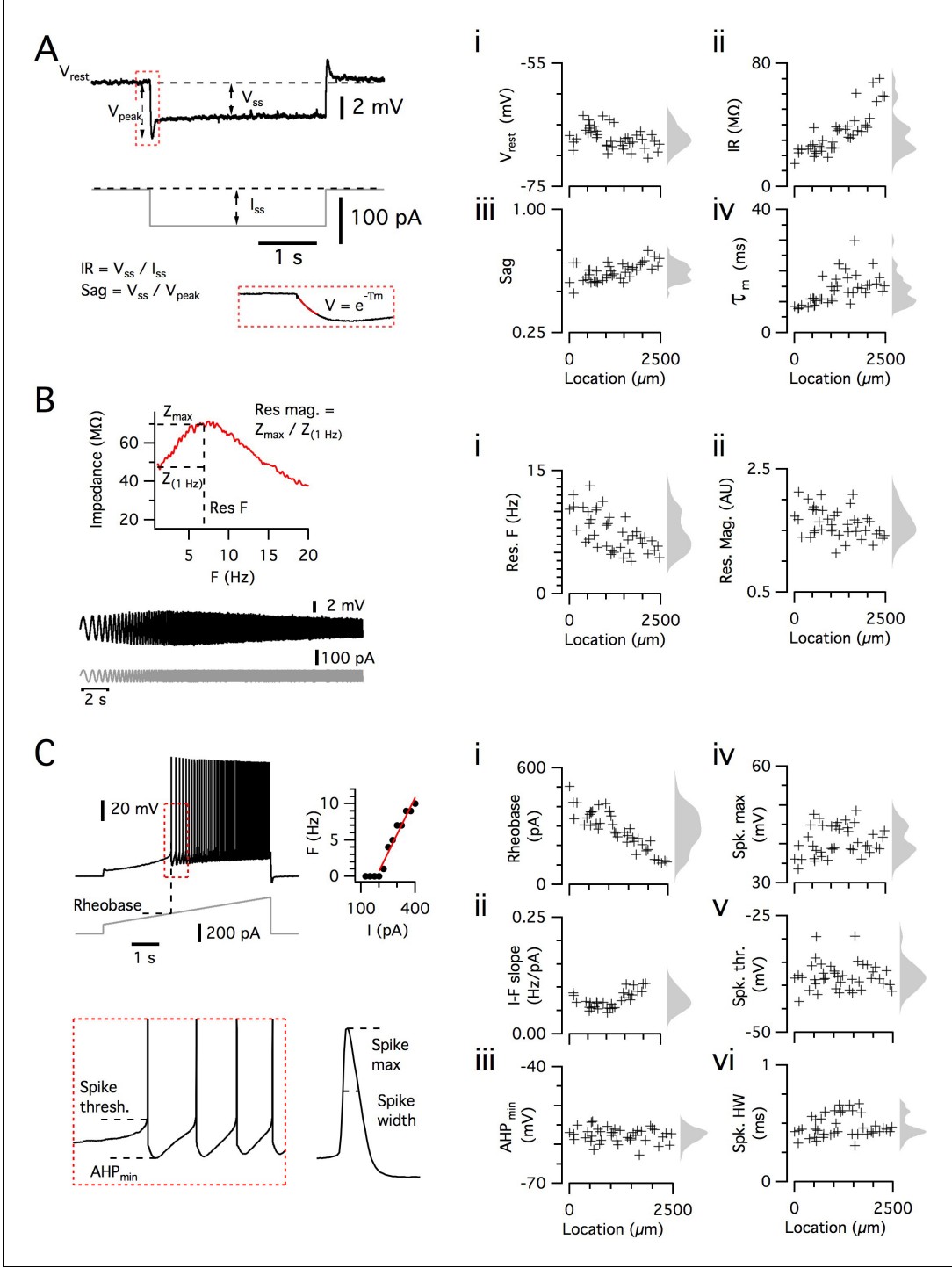

**Figure 2.** Dorsoventral organization of intrinsic properties of stellate cells from a single animal. (**A–C**) Waveforms (gray traces) and example responses (black traces) from a single mouse for step (**A**), ZAP (**B**) and ramp (**C**) stimuli (left). The properties derived from each protocol are shown plotted against recording location (each data point is a black cross) (right). KSDs with arbitrary bandwidth are displayed to the right of each data plot to facilitate visualization of the property's distribution when location information is disregarded (light gray pdfs). (**A**) Injection of a series of current steps enables the measurement of the resting membrane potential ($V_{rest}$) (**i**), the input resistance (IR) (**ii**), the sag coefficient (sag) (**iii**) and the membrane time constant ($\tau_m$) (**iv**). (**B**) Injection of ZAP current waveform enables the calculation of an impedance amplitude profile, which was used to estimate the resonance resonant frequency (Res. F) (**i**) and magnitude (Res. mag) (**ii**). (**C**) Injection of a slow current ramp enabled

*Figure 2 continued on next page*

*Figure 2 continued*

the measurement of the rheobase (i); the slope of the current-frequency relationship (I-F slope) (ii); using only the first five spikes in each response (enlarged zoom, lower left), the AHP minimum value (AHP$_{min}$) (iii); the spike maximum (Spk. max) (iv); the spike threshold (Spk. thr.) (v); and the spike width at half height (Spk. HW) (vi).

The online version of this article includes the following figure supplement(s) for figure 2:

**Figure supplement 1.** Large environment for housing.

If set points that determine integrative properties of SCs do indeed differ between animals, then mixed models should provide a better account of the data than linear models that are generated by pooling data across all animals. Consistent with this, we found that mixed models for all electrophysiological features gave a substantially better fit to the data than linear models that considered all neurons as independent (adjusted $p < 2 \times 10^{-17}$ for all models, $\chi^2$ test, *Table 1*). Furthermore, even for properties with substantial ($R^2$ value $> 0.1$) dorsoventral tuning, the conditional $R^2$ value for the mixed effect model was substantially larger than the marginal $R^2$ value (*Figure 4D* and *Table 1*). Together, these analyses demonstrate inter-animal variability in key electrophysiological features of SCs, suggesting that the set points that establish the underlying integrative properties differ between animals.

## Experience-dependence of intrinsic properties of stellate cells

Because neuronal integrative properties may be modified by changes in neural activity (*Zhang and Linden, 2003*), we asked whether experience influences the measured electrophysiological features of SCs. We reasoned that modifying the space through which animals can navigate may drive experience-dependent plasticity in the MEC. As standard mouse housing has dimensions less than the distance between the firing fields of more ventrally located grid cells (*Brun et al., 2008*; *Hafting et al., 2005*), in a standard home cage, only a relatively small fraction of ventral grid cells is likely to be activated, whereas larger housing should lead to the activation of a greater proportion of ventral grid cells. We therefore tested whether the electrophysiological features of SCs differ between mice housed in larger environments (28,800 cm$^2$) and those with standard home cages (740 cm$^2$).

We compared the mixed models described above to models in which housing was also included as a fixed effect. To minimize the effects of age on SCs (*Boehlen et al., 2010*; *Burton et al., 2008*; *Supplementary file 2*), we focused these and subsequent analyses on mice between P33 and P44 (N = 25, n = 779). We found that larger housing was associated with a smaller sag coefficient, indicating an increased sag response, a lower resonant frequency and a larger spike half-width (adjusted $p < 0.05$; *Figure 4E*, *Supplementary file 3*). These differences were primarily from changes to the magnitude rather than the location-dependence of each feature. Other electrophysiological features appeared to be unaffected by housing.

To determine whether inter-animal differences remain after accounting for housing, we compared mixed models that include dorsoventral location and housing as fixed effects with equivalent linear regression models in which individual animals were not accounted for. Mixed models incorporating animal identity continued to provide a better account of the data, both for features that were dependent on housing (adjusted $p < 2.8 \times 10^{-21}$) and for features that were not (adjusted $p < 1.4 \times 10^{-7}$) (*Supplementary file 4*).

Together, these data suggest that specific electrophysiological features of SCs may be modified by experience of large environments. After accounting for housing, significant inter-animal variation remains, suggesting that additional mechanisms acting at the level of animals rather than individual neurons also determine differences between SCs.

## Inter-animal differences remain after accounting for additional experimental parameters

To address the possibility that other experimental or biological variables could contribute to inter-animal differences, we evaluated the effects of home cage size (*Supplementary files 3–4*), brain hemisphere (*Supplementary file 5*), mediolateral position (*Figure 4—figure supplement 1* and *Supplementary file 6*), the identity of the experimenter (*Supplementary file 7*) and time since slice preparation (*Supplementary files 8* and *9*). Several of the variables influenced some measured

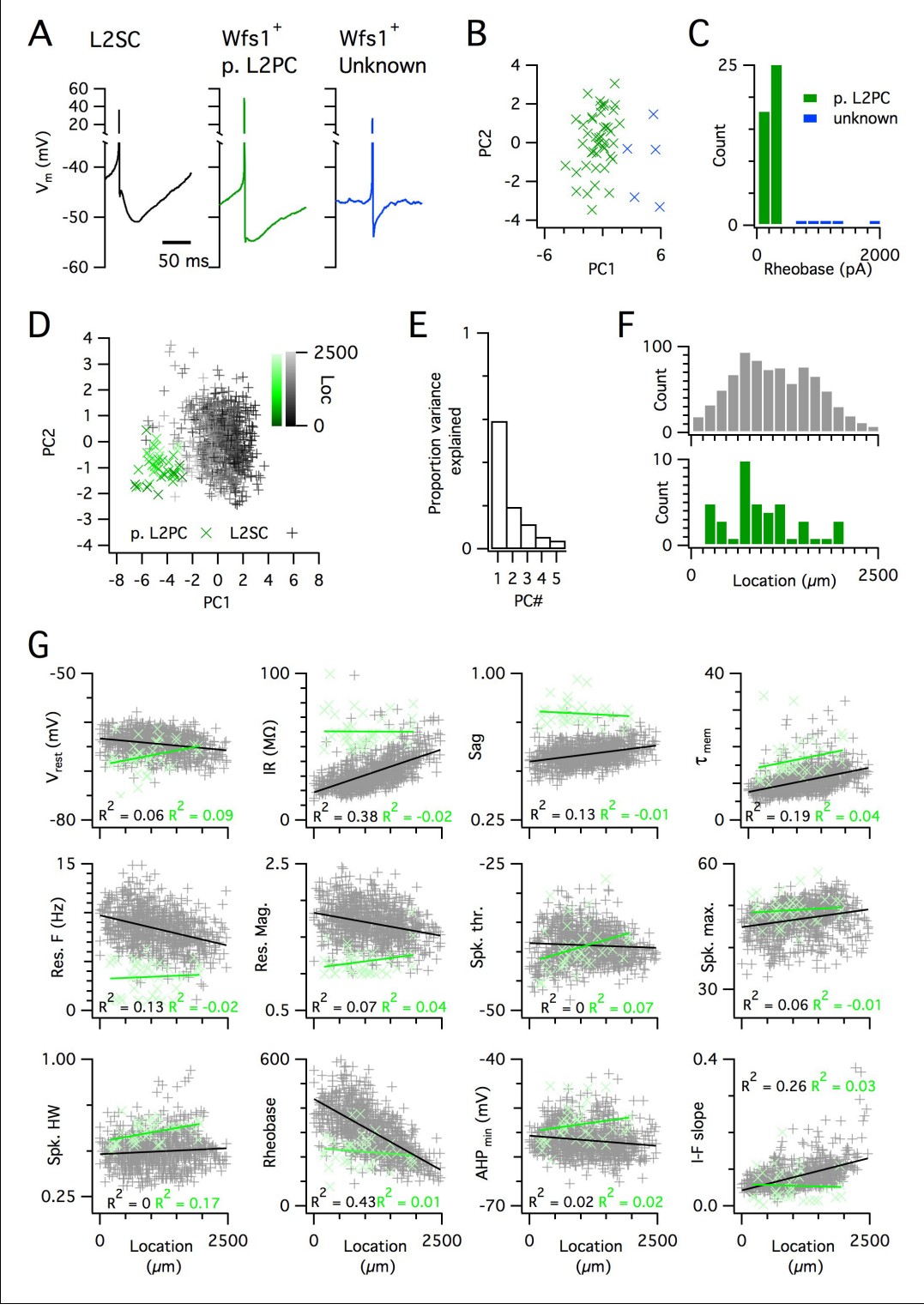

**Figure 3.** Distinct and dorsoventrally organized properties of layer 2 stellate cells. (A) Representative action potential after hyperpolarization waveforms from a SC (left), a pyramidal cell (middle) and an unidentified cell (right). The pyramidal and unidentified cells were both positively labelled in *Wfs1*[Cre] mice. (B) Plot of the first versus the second principal component from PCA of the properties of labelled neurons in *Wfs1*[Cre] mice reveals two populations of neurons. (C) Histogram showing the distribution of rheobase values of cells positively labelled in *Wfs1*[Cre] mice. The two groups identified in panel (B) can be distinguished by their rheobase. (D) Plot of the first

*Figure 3 continued on next page*

two principal components from PCA of the properties of the L2PC (n = 44, green) and SC populations (n = 836, black). Putative pyramidal cells (x) and SCs (+) are colored according to their dorsoventral location (inset shows the scale). (E) Proportion of total variance explained by the first five principal components for the analysis in panel (D). (F) Histograms of the locations of recorded SCs (upper) and L2PCs (lower). (G) All values of measured features from all mice are plotted as a function of the dorsoventral location of the recorded cells. Lines indicate fits of a linear model to the complete datasets for SCs (black) and L2PCs (green). Putative pyramidal cells (x, green) and SCs (+, black). Adjusted R$^2$ values use the same color scheme.

electrophysiological features, for example properties primarily related to the action potential wave-form depended on the mediolateral position of the recorded neuron (*Supplementary file 6*; *Canto and Witter, 2012*; *Yoshida et al., 2013*), but significant inter-animal differences remained after accounting for each variable. We carried out further analyses using models that included housing, mediolateral position, experimenter identity and the direction in which sequential recordings were obtained as fixed effects (*Supplementary file 10*), and using models fit to minimal datasets in which housing, mediolateral position and the recording direction were identical (*Supplementary file 11*). These analyses again found evidence for significant inter-animal differences.

Inter-animal differences could arise if the health of the recorded neurons differed between brain slices. To minimize this possibility, we standardized our procedures for tissue preparation (see 'Materials and methods'), such that slices were of consistent high quality as assessed by low numbers of unhealthy cells and by visualization of soma and dendrites of neurons in the slice. Several further

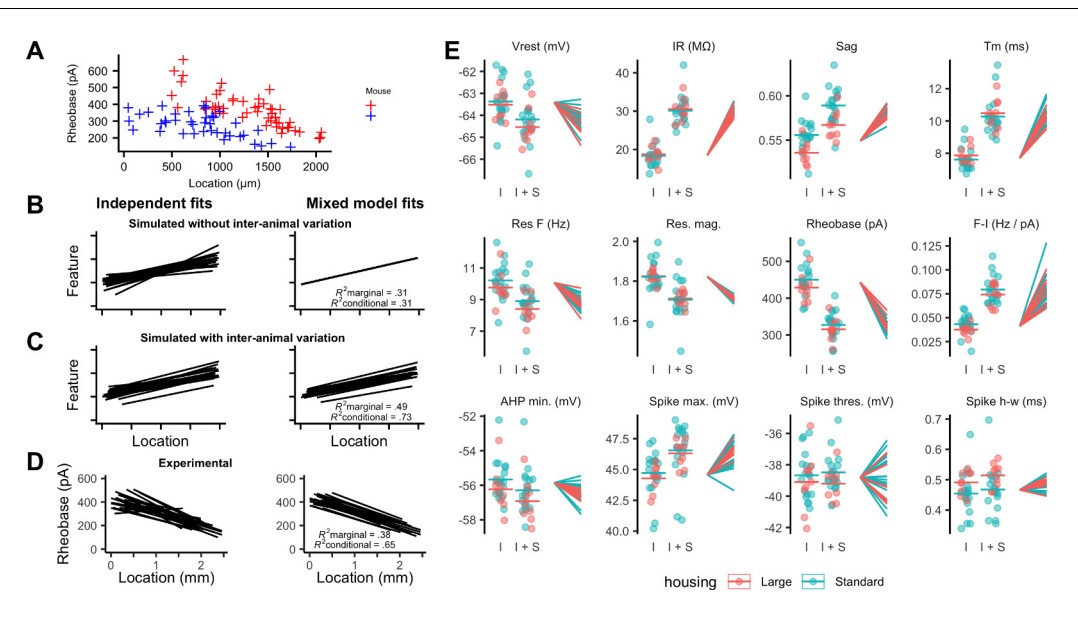

**Figure 4.** Inter-animal variability and dependence on environment of intrinsic properties of stellate cells. (A) Examples of rheobase as a function of dorsoventral position for two mice. (B, C) Each line is the fit of simulated data from a different subject for datasets in which there is no inter-subject variability (B) or in which intersubject variability is present (C). Fitting data from each subject independently with linear regression models suggests intersubject variation in both datasets (left). By contrast, after fitting mixed effect models (right) intersubject variation is no longer suggested for the dataset in which it is absent (B) but remains for the dataset in which it is present (C). (D) Each line is the fit of rheobase as a function of dorsoventral location for a single mouse. The fits were carried out independently for each mouse (left) or using a mixed effect model with mouse identity as a random effect (right). (E) The intercept (I), sum of the intercept and slope (I + S), and slopes realigned to a common intercept (right) for each mouse obtained by fitting mixed effect models for each property as a function of dorsoventral position.

The online version of this article includes the following figure supplement(s) for figure 4:

**Figure supplement 1.** Properties of SCs in medial and lateral slices.

**Table 1.** Dependence of the electrophysiological features of SCs on dorsoventral position.

Key statistical parameters from analyses of the relationship between each measured electrophysiological feature and dorsoventral location. The data are ordered according to the marginal $R^2$ for each property's relationship with dorsoventral position. Slope is the population slope from fitting a mixed effect model for each feature with location as a fixed effect ($mm^{-1}$), with p(slope) obtained by comparing this model to a model without location as a fixed effect ($\chi^2$ test). For all properties except the spike threshold, the slope was unlikely to have arisen by chance ($p<0.05$). The marginal and conditional $R^2$ values, and the minimum and maximum slopes across all mice, are obtained from the fits of mixed effect models that contain location as a fixed effect. The estimate p(vs linear) is obtained by comparing the mixed effect model containing location as a fixed effect with a corresponding linear model without random effects ($\chi^2$ test). The values of p(slope) and p(vs linear) were adjusted for multiple comparisons using the method of *Benjamini and Hochberg (1995)*. Units for the slope measurements are units for the property $mm^{-1}$. The data are from 27 mice.

| Feature | Slope | P (slope) | Marginal $R^2$ | Conditional $R^2$ | Slope (min) | Slope (max) | P (vs linear) |
|---|---|---|---|---|---|---|---|
| IR (MΩ) | 11.794 | 8.39e-17 | 0.383 | 0.532 | 9.630 | 14.262 | 4.33e-40 |
| Rheobase (pA) | −119.887 | 9.07e-15 | 0.382 | 0.652 | −153.873 | −76.130 | 6.55e-43 |
| I-F slope (Hz/pA) | 0.036 | 6.06e-10 | 0.228 | 0.561 | 0.019 | 0.087 | 6.82e-34 |
| Tm (ms) | 2.646 | 3.70e-12 | 0.192 | 0.343 | 1.809 | 3.979 | 1.20e-29 |
| Res. frequency (Hz) | −1.334 | 4.13e-09 | 0.122 | 0.553 | −2.299 | −0.342 | 6.37e-65 |
| Sag | 0.033 | 6.06e-10 | 0.121 | 0.347 | 0.016 | 0.043 | 1.91e-38 |
| Spike maximum (mV) | 1.900 | 1.85e-05 | 0.064 | 0.436 | −1.288 | 3.297 | 1.14e-50 |
| Res. magnitude | −0.114 | 6.34e-08 | 0.064 | 0.198 | −0.138 | −0.087 | 9.13e-20 |
| Vm (mV) | −0.884 | 3.67e-05 | 0.046 | 0.348 | −1.965 | 0.150 | 8.73e-35 |
| Spike AHP (mV) | −0.645 | 1.93e-02 | 0.011 | 0.257 | −1.828 | 0.408 | 1.82e-17 |
| Spike width (ms) | 0.017 | 1.93e-02 | 0.010 | 0.643 | −0.021 | 0.055 | 7.04e-139 |
| Spike threshold (mV) | 0.082 | 8.20e-01 | 0.000 | 0.510 | −2.468 | 2.380 | 2.03e-17 |

observations are consistent with comparable quality of slices between experiments. First, if the condition of the slices had differed substantially between animals, then in better quality slices, it should be easier to record from more neurons, in which case features that depend on tissue quality would correlate with the number of recorded neurons. However, the majority (10/12) of the electrophysiological features were not significantly ($p>0.2$) associated with the number of recorded neurons (*Supplementary file 12*). Second, analyses of inter-animal differences that focus only on data from animals for which >35 recordings were made, which should only be feasible with uniformly high-quality brain slices, are consistent with conclusions from analysis of the larger dataset (*Supplementary file 13*). Third, the conditional $R^2$ values of electrophysiological features of L2PCs are much lower than those for SCs recorded under the same experimental conditions (*Table 1* and *Supplementary file 1*), suggesting that inter-animal variation may be specific to SCs and cannot be explained by slice conditions. Together, these analyses indicate that differences between animals remain after accounting for experimental and technical factors that might contribute to variation in the measured features of SCs.

## The distribution of intrinsic properties is consistent with a continuous rather than a modular organization

The dorsoventral organization of SC integrative properties is well established, but whether this results from within animal variation consistent with a small number of discrete set points that underlie a modular organization (*Figure 1B*) is unclear. To evaluate modularity, we used datasets with $n \geq 34$ SCs (N = 15 mice, median age = 37 days, age range = 18–43 days). We focus initially on rheobase, which is the property with the strongest correlation to dorsoventral location, and resonant frequency, which is related to the oscillatory dynamics underlying dorsoventral tuning in some models of grid firing (e.g. *Burgess et al., 2007*; *Giocomo et al., 2007*). For $n \geq 34$ SCs, we expect that if properties are modular, then this would be detected by the modified gap statistic in at least 50% of animals (*Figure 1—figure supplements 2C* and *3*). By contrast, we find that for datasets from the majority of animals, the modified gap statistic identifies only a single mode in the distribution of

rheobase values (*Figure 5A* and *Figure 6*) (N = 13/15) and of resonant frequencies (*Figure 5B* and *Figure 6*) (N = 14/15), indicating that these properties have a continuous rather than a modular distribution. Consistent with this, smoothed distributions did not show clearly separated peaks for either property (*Figure 5*). The mean and 95% confidence interval for the probability of evaluating a dataset as clustered ($p_{detect}$) was 0.133 and 0.02–0.4 for rheobase and 0.067 and 0.002–0.32 for resonant frequency. These values of $p_{detect}$ were not significantly different from the proportions expected given the false positive rate of 0.1 in the complete absence of clustering (p=0.28 and 0.66,

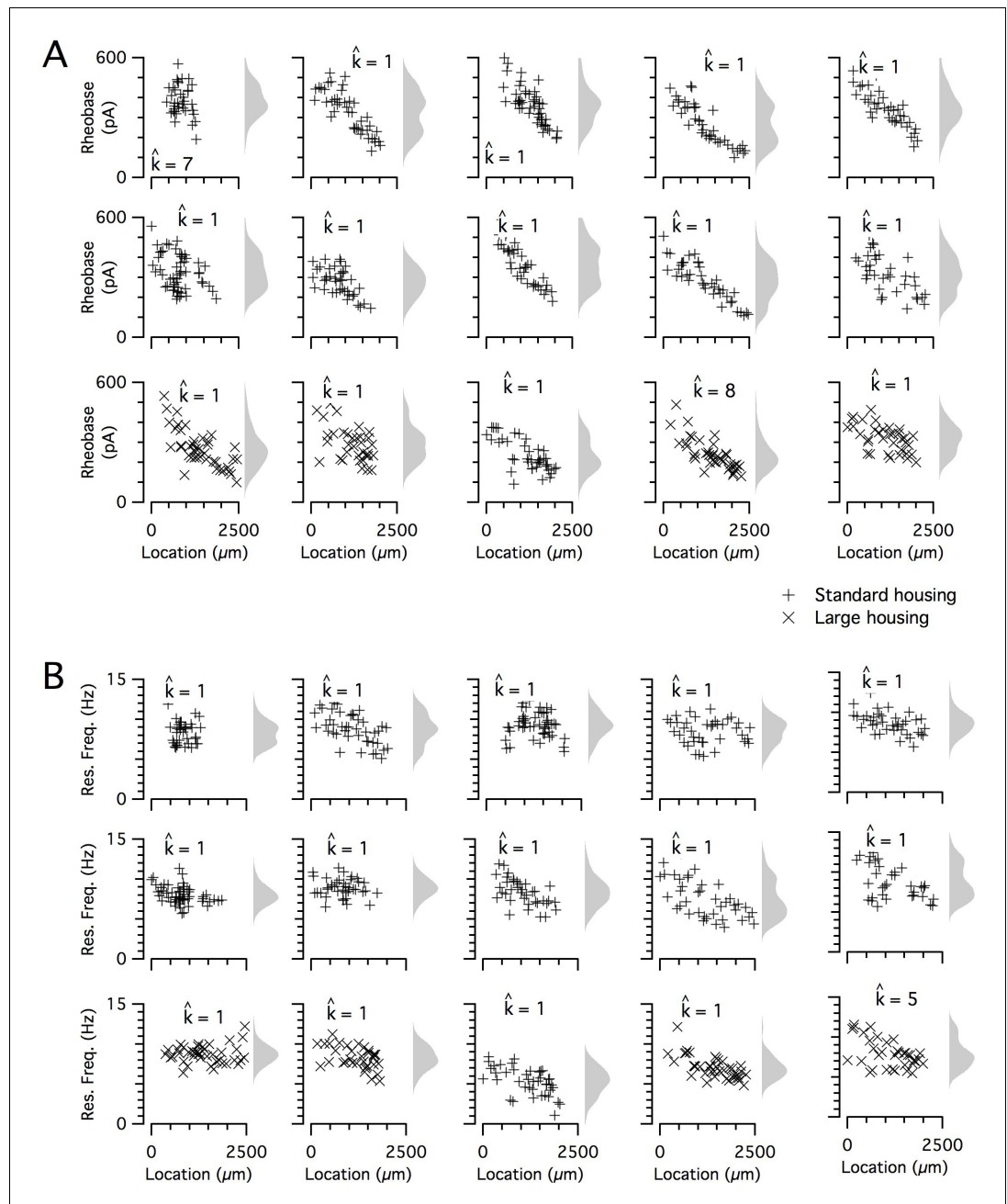

**Figure 5.** Rheobase and resonant frequency do not have a detectable modular organization. (A, B) Rheobase (A) and resonant frequency (B) are plotted as a function of dorsoventral position separately for each animal. Marker color and type indicate the age and housing conditions of the animal ('+' standard housing, 'x' large housing). KSDs (arbitrary bandwidth, same for all animals) are also shown. The number of clusters in the data ($k_{est}$) is indicated for each animal ($\hat{k}$).

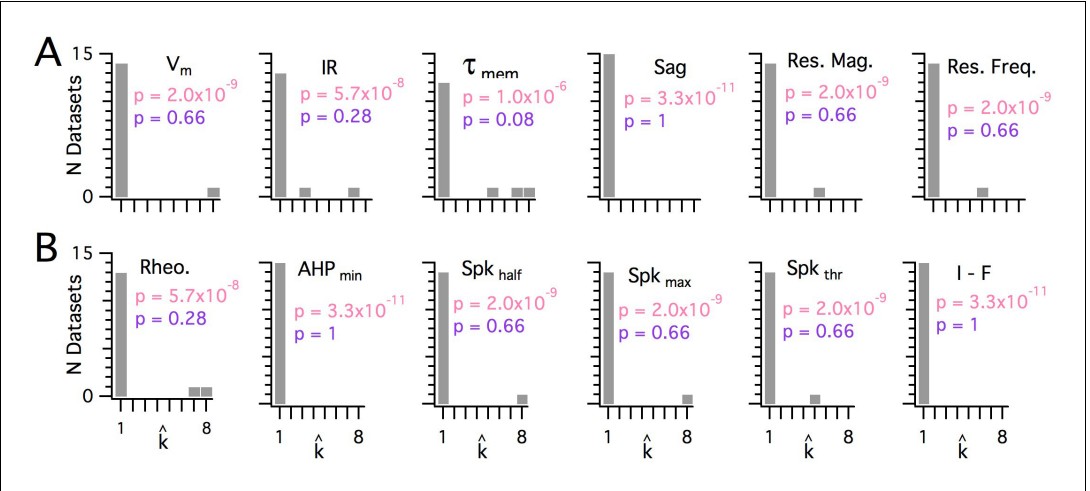

**Figure 6.** Significant modularity is not detectable for any measured property. (A, B) Histograms showing the $k_{est}$ ($\hat{k}$) counts across all mice for each different measured sub-threshold (A) and supra-threshold (B) intrinsic property. The maximum k evaluated was 8. The proportion of each measured property with $k_{est}>1$ was compared using binomial tests (with N = 15) to the proportion expected if the underlying distribution of that property is always clustered with all separation between modes $\geq 5$ standard deviations (pink text), or if the underlying distribution of the property is uniform (purple text). For all measured properties, the values of $k_{est}$ ($\hat{k}$) were indistinguishable (p>0.05) from the data generated from a uniform distribution and differed from the data generated from a multi-modal distribution ($p<1\times10^{-6}$).

binomial test). Thus, the rheobase and resonant frequency of SCs, although depending strongly on a neuron's dorsoventral position, do not have a detectable modular organization.

When we investigated the other measured integrative properties, we also failed to find evidence for modularity. Across all properties, for any given property, at most 3 out of 15 mice were evaluated as having a clustered organization using the modified gap statistic (*Figure 6*). This does not differ significantly from the proportion expected by chance when no modularity is present (p>0.05, binomial test). Consistent with this, the average proportion of datasets evaluated as modular across all measured features was 0.072 ± 0.02 (± SEM), which is similar to the expected false-positive rate. By contrast, the properties of grid firing fields previously recorded with tetrodes in behaving animals (*Stensola et al., 2012*) were detected as having a modular organization using the modified gap statistic (*Figure 1—figure supplement 5*). For seven grid-cell datasets with n $\geq$ 20, the mean for $p_{detect}$ is 0.86, with 95% confidence intervals of 0.42 to 0.996. We note here that discontinuity algorithms that were previously used to assess the modularity of grid field properties (*Giocomo et al., 2014*; *Stensola et al., 2012*) did indicate significant modularity in the majority of the intrinsic properties measured in our dataset (N = 13/15 and N = 12/15, respectively), but this was attributable to false positives resulting from the relatively even sampling of recording locations (see *Figure 1—figure supplement 4A*). Therefore, we conclude that it is unlikely that any of the intrinsic integrative properties of SCs that we examined have organization within individual animals resembling the modular organization of grid cells in behaving animals.

## Multiple sources of variance contribute to diversity in stellate cell intrinsic properties

Finally, because many of the measured electrophysiological features of SCs emerge from shared ionic mechanisms (*Dodson et al., 2011*; *Garden et al., 2008*; *Pastoll et al., 2012a*), we asked whether dorsoventral tuning reflects a single core mechanism and whether inter-animal differences are specific to this mechanism or manifest more generally.

Estimation of conditional independence for measurements at the level of individual neurons (*Figure 7A*) or individual animals (*Figure 7B*) was consistent with the expectation that particular classes of membrane ion channels influence multiple electrophysiologically measured features. The first five dimensions of a principal components analysis (PCA) of all measured electrophysiological

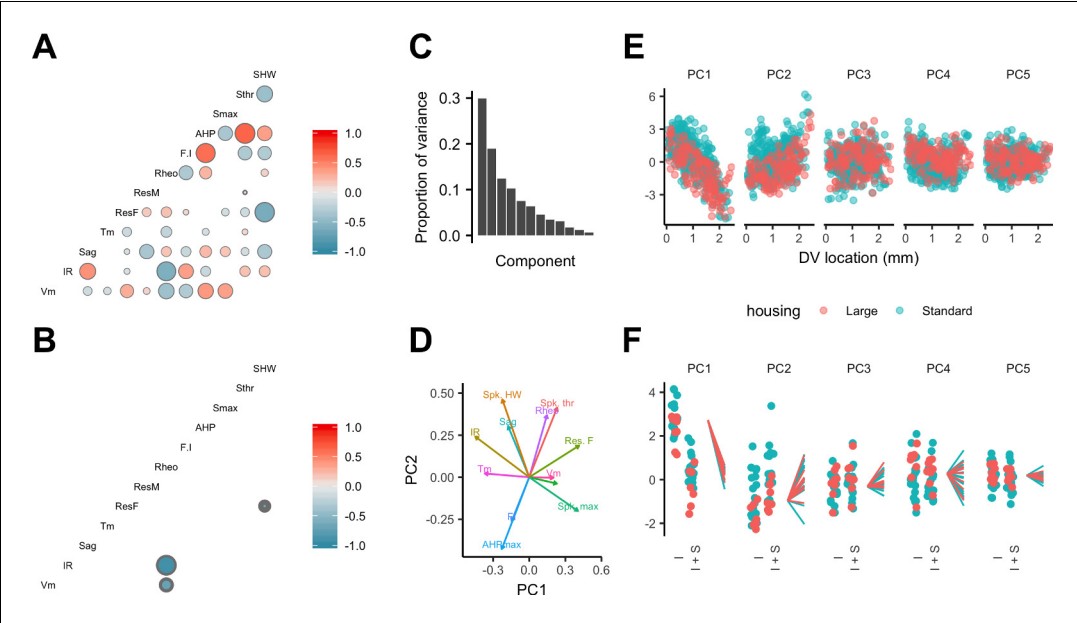

**Figure 7.** Feature relationships and inter-animal variability after reducing dimensionality of the data. (A, B) Partial correlations between the electrophysiological features investigated at the level of individual neurons (A) and at the level of animals (B). Partial correlations outside of the 95% basic bootstrap confidence intervals are color coded. Non-significant correlations are colored white. Properties shown are the resting membrane potential (Vm), input resistance (IR), membrane potential sag response (sag), membrane time constant (Tm), resonance frequency (Rm), resonance magnitude (Rm), rheobase (Rheo), slope of the current frequency relationship (FI), peak of the action potential after hyperpolarization (AHP), peak of the action potential (Smax) voltage threshold for the action potential (Sthr) and half-width of the action potential (SHW). (C) Proportion of variance explained by each principal component. To remove variance caused by animal age and the experimenter identity, the principal component analysis used a reduced dataset obtained by one experimenter and restricted to animals between 32 and 45 days old (N = 25, n = 572). (D) Loading plot for the first two principal components. (E) The first five principal components plotted as a function of position. (F) Intercept (I), intercept plus the slope (I + S) and slopes aligned to the same intercept, for fits for each animal of the first five principal components to a mixed model with location as a fixed effect and animal as a random effect.

features accounted for almost 80% of the variance (*Figure 7C*). Examination of the rotations used to generate the principal components suggested relationships between individual features that are consistent with our evaluation of the conditional independence structure of the measured features (*Figure 7D and A*). When we fit the principal components using mixed models with location as a fixed effect and animal identity as a random effect, we found that the first two components depended significantly on dorsoventral location (*Figure 7E* and *Supplementary file 14*) (marginal $R^2$ = 0.50 and 0.09 and adjusted p=$1.09 \times 10^{-15}$ and $1.05 \times 10^{-4}$, respectively). Thus, the dependence of multiple electrophysiological features on dorsoventral position may be reducible to two core mechanisms that together account for much of the variability between SCs in their intrinsic electrophysiology.

Is inter-animal variation present in PCA dimensions that account for dorsoventral variation? The intercept, but not the slope of the dependence of the first two principal components on dorsoventral position depended on housing (adjusted p=0.039 and 0.027) (*Figure 7E,F* and *Supplementary file 15*). After accounting for housing, the first two principal components were still better fit by models that include animal identity as a random effect (adjusted p=$3.3 \times 10^{-9}$ and $4.1 \times 10^{-86}$), indicating remaining inter-animal differences in these components (*Supplementary file 16*). A further nine of the next ten higher-order principal components did not depend on housing (adjusted p>0.1) (*Supplementary file 15*), while eight differed significantly between animals (adjusted p<0.05) (*Supplementary file 16*).

Together, these analyses indicate that the dorsoventral organization of multiple electrophysiological features of SCs is captured by two principal components, suggesting two main sources of variance, both of which are dependent on experience. Significant inter-animal variation in the major sources of variance remains after accounting for experience and experimental parameters.

## Discussion

Phenotypic variation is found across many areas of biology (*Geiler-Samerotte et al., 2013*), but has received little attention in investigations of mammalian nervous systems. We find unexpected inter-animal variability in SC properties, suggesting that the integrative properties of neurons are determined by set points that differ between animals and are controlled at a circuit level (*Figure 8*). Continuous, location-dependent organization of set points for SC integrative properties provides new constraints on models for grid cell firing. More generally, the existence of inter-animal differences in set points has implications for experimental design and raises new questions about how the integrative properties of neurons are specified.

### A conceptual framework for within cell type variability

Theoretical models suggest how different cell types can be generated by varying target concentrations of intracellular $Ca^{2+}$ or rates of ion channel expression (*O'Leary et al., 2014*). The within cell type variability predicted by these models arises from different initial conditions and may explain the variability in our data between neurons from the same animal at the same dorsoventral location (*Figure 8A*). By contrast, the dependence of integrative properties on position and their variation between animals implies additional mechanisms that operate at the circuit level (*Figure 8B*). In principle, this variation could be accounted for by inter-animal differences in dorsoventrally tuned or

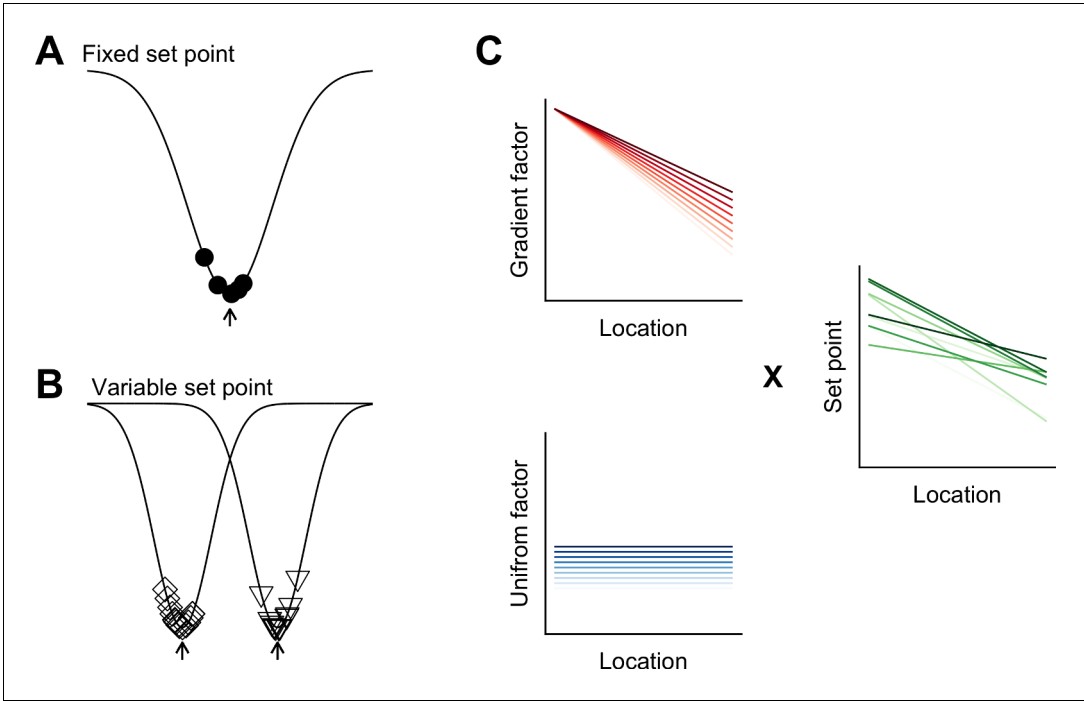

**Figure 8.** Models for intra- and inter-animal variation. (A) The configuration of a cell type can be conceived of as a trough (arrow head) in a developmental landscape (solid line). In this scheme, the trough can be considered as a set point. Differences between cells (filled circles) reflect variation away from the set point. (B) Neurons from different animals or located at different dorsoventral positions can be conceptualized as arising from different troughs in the developmental landscape. (C) Variation may reflect inter-animal differences in factors that establish gradients (upper left) and in factors that are uniformly distributed (lower left), combining to generate set points that depend on animal identity and location (right). Each line corresponds to schematized properties of a single animal.

spatially uniform factors that influence ion channel expression or target points for intracellular Ca$^{2+}$ (*Figure 8C*).

The mechanisms for within cell type variability that are suggested by our results may differ from inter-animal variation described in invertebrate nervous systems. In invertebrates, inter-animal variability is between properties of individual identified neurons (*Goaillard et al., 2009*), whereas in mammalian nervous systems, neurons are not individually identifiable and the variation that we describe here is at the level of cell populations. From a developmental perspective in which cell identity is considered as a trough in a state-landscape through which each cell moves (*Wang et al., 2011b*), variation in the population of neurons of the same type could be conceived as cell autonomous deviations from set points corresponding to the trough (*Figure 8A*). Our finding that variability among neurons of the same type manifests between as well as within animals, could be explained by differences between animals in the position of the trough or set point in the developmental landscape (*Figure 8B*).

Our comparison of neurons from animals in standard and large cages provides evidence for the idea that within cell-type excitable properties are modified by experience (*Zhang and Linden, 2003*). For example, granule cells in the dentate gyrus that receive input from SCs increase their excitability when animals are housed in enriched environments (*Green and Greenough, 1986*; *Ohline and Abraham, 2019*). Our experiments differ in that we increased the size of the environment with the goal of activating more ventral grid cells, whereas previous enrichment experiments have focused on increasing the environmental complexity and availability of objects for exploration. Further investigation will be required to dissociate the influence of each factor on excitability.

## Implications of continuous dorsoventral organization of stellate cell integrative properties for grid cell firing

Dorsoventral gradients in the electrophysiological features of SCs have stimulated cellular models for the organization of grid firing (*Burgess, 2008*; *Giocomo and Hasselmo, 2008b*; *Grossberg and Pilly, 2012*; *O'Donnell and Nolan, 2011*; *Widloski and Fiete, 2014*). Increases in spatial scale following deletion of HCN1 channels (*Giocomo et al., 2011*), which in part determine the dorsoventral organization of SC integrative properties (*Garden et al., 2008*; *Giocomo and Hasselmo, 2009*), support a relationship between the electrophysiological properties of SCs and grid cell spatial scales. Our data argue against models that explain this relationship through single cell computations (*Burgess, 2008*; *Burgess et al., 2007*; *Giocomo et al., 2007*), as in this case, the modularity of integrative properties of SCs is required to generate modularity of grid firing. A continuous dorsoventral organization of the electrophysiological properties of SCs could support the modular grid firing generated by self-organizing maps (*Grossberg and Pilly, 2012*) or by synaptic learning mechanisms (*Kropff and Treves, 2008*; *Urdapilleta et al., 2017*). It is less clear how a continuous gradient would affect the organization of grid firing predicted by continuous attractor network models, which can instead account for modularity by limiting synaptic interactions between modules (*Burak and Fiete, 2009*; *Bush and Burgess, 2014*; *Fuhs and Touretzky, 2006*; *Guanella et al., 2007*; *Shipston-Sharman et al., 2016*; *Widloski and Fiete, 2014*; *Yoon et al., 2013*). Modularity of grid cell firing could also arise through the anatomical clustering of calbindin-positive L2PCs (*Ray et al., 2014*; *Ray and Brecht, 2016*). Because many SCs do not appear to generate grid codes and as the most abundant functional cell type in the MEC appears to be non-grid spatial neurons (*Diehl et al., 2017*; *Hardcastle et al., 2017*), the continuous dorsoventral organization of SC integrative properties may also impact grid firing indirectly through modulation of these codes.

Our results add to previous comparisons of medially and laterally located SCs (*Canto and Witter, 2012*; *Yoshida et al., 2013*). The similar dorsoventral organization of subthreshold integrative properties of SCs from medial and lateral parts of the MEC appears consistent with the organization of grid cell modules recorded in behaving animals (*Stensola et al., 2012*). How mediolateral differences in firing properties (*Figure 4—figure supplement 1*; *Canto and Witter, 2012*; *Yoshida et al., 2013*) might contribute to spatial computations within the MEC is unclear.

The continuous dorsoventral variation of the electrophysiological features of SCs suggested by our analysis is consistent with continuous dorsoventral gradients in gene expression along layer 2 of the MEC (*Ramsden et al., 2015*). For example, labelling of the mRNA and protein for the HCN1 ion channel suggests a continuous dorsoventral gradient in its expression (*Nolan et al., 2007*; *Ramsden et al., 2015*). It is also consistent with single-cell RNA sequencing analysis of other brain

areas, which indicates that although the expression profiles for some cell types cluster around a point in feature space, others lie along a continuum (*Cembrowski and Menon, 2018*). It will be interesting in future to determine whether gene expression continua establish corresponding continua of electrophysiological features (*Liss et al., 2001*).

### Functional consequences of within cell type inter-animal variability

What are the functional roles of inter-animal variability? In the crab stomatogastric ganglion, inter-animal variation correlates with circuit performance (*Goaillard et al., 2009*). Accordingly, variation in intrinsic properties of SCs might correlate with differences in grid firing (*Domnisoru et al., 2013*; *Gu et al., 2018*; *Rowland et al., 2018*; *Schmidt-Hieber and Häusser, 2013*) or behaviors that rely on SCs (*Kitamura et al., 2014*; *Qin et al., 2018*; *Tennant et al., 2018*). It is interesting in this respect that there appear to be inter-animal differences in the spatial scale of grid modules (Figure 5 of *Stensola et al., 2012*). Modification of grid field scaling following deletion of HCN1 channels is also consistent with this possibility (*Giocomo et al., 2011*; *Mallory et al., 2018*). Alternatively, inter-animal differences may reflect multiple ways to achieve a common higher-order phenotype. According to this view, coding of spatial location by SCs would not differ between animals despite lower level variation in their intrinsic electrophysiological features. This is related to the idea of degeneracy at the level of single-cell electrophysiological properties (*Marder and Goaillard, 2006*; *Mittal and Narayanan, 2018*; *O'Leary et al., 2014*; *Swensen and Bean, 2005*), except that here the electrophysiological features differ between animals whereas the higher-order circuit computations may nevertheless be similar.

In conclusion, our results identify substantial within cell type variation in neuronal integrative properties that manifests between as well as within animals. This has implications for experimental design and model building as the distribution of replicates from the same animal will differ from those obtained from different animals (*Marder and Taylor, 2011*). An important future goal will be to distinguish causes of inter-animal variation. Many behaviors are characterized by substantial inter-animal variation (e.g. *Villette et al., 2017*), which could result from variation in neuronal integrative properties, or could drive this variation. For example, it is possible that external factors such as social interactions may affect brain circuitry (*Wang et al., 2011a*; *Wang et al., 2014*), although these effects appear to be focused on frontal cortical structures rather than circuits for spatial computations (*Wang et al., 2014*). Alternatively, stochastic mechanisms operating at the population level may drive the emergence of inter-animal differences during the development of SCs (*Donato et al., 2017*; *Ray and Brecht, 2016*). Addressing these questions may turn out to be critical to understanding the relationship between cellular biophysics and circuit-level computations in cognitive circuits (*Schmidt-Hieber and Nolan, 2017*).

## Materials and methods

### Mouse strains

All experimental procedures were performed under a United Kingdom Home Office license and with approval of the University of Edinburgh's animal welfare committee. Recordings of many SCs per animal used C57Bl/6J mice (Charles River). Recordings targeting calbindin cells used a *Wfs1*[Cre] line (*Wfs1*-Tg3-CreERT2) obtained from Jackson Labs (Strain name: B6;C3-Tg(*Wfs1*-cre/ERT2)3Aibs/J; stock number:009103) crossed to RCE:loxP (R26R CAG-boosted EGFP) reporter mice (described in *Miyoshi et al., 2010*). To promote expression of Cre in the mice, tamoxifen (Sigma, 20 mg/ml in corn oil) was administered on three consecutive days by intraperitoneal injections, approximately 1 week before experiments. Mice were group housed in a 12 hr light/dark cycle with unrestricted access to food and water (light on 07.30–19.30 hr). Mice were usually housed in standard $0.2 \times 0.37$ m cages that contained a cardboard roll for enrichment. A subset of the mice was instead housed from pre-weaning ages in a larger $2.4 \times 1.2$ m cage that was enriched with up to 15 bright plastic objects and eight cardboard rolls (*Figure 2—figure supplement 1*). Thus, the large cages had more items but at a slightly lower density (large cages — up to 1 item per 0.125 m$^2$; standard cages — 1 item per 0.074 m$^2$). All experiments in the standard cage used male mice. For experiments in the large cage, two mice were female, six mice were male and one was not identified. Because we did not find significant effects of sex on individual electrophysiologically properties,

all mice were included in the analyses reported in the text. When only male mice were included, the effects of housing on the first principal component remained significant, whereas the effects of housing on individual electrophysiologically properties no longer reach statistical significance after correcting for multiple comparisons. Additional analyses that consider only male mice are provided in the code associated with the manuscript.

## Slice preparation

Methods for preparation of parasagittal brain slices containing medial entorhinal cortex were based on procedures described previously (*Pastoll et al., 2012b*). Briefly, mice were sacrificed by cervical dislocation and their brains carefully removed and placed in cold (2–4°C) modified ACSF, with composition (in mM): NaCl 86, $NaH_2PO_4$ 1.2, KCl 2.5, $NaHCO_3$ 25, glucose 25, sucrose 75, $CaCl_2$ 0.5, and $MgCl_2$ 7. Brains were then hemisected and sectioned, also in modified ACSF at 4–8°C, using a vibratome (Leica VT1200S). To minimize variation in the slicing angle, the hemi-section was cut along the midline of the brain and the cut surface of the brain was glued to the cutting block. After cutting, brains were placed at 36°C for 30 min in standard ACSF, with composition (in mM): NaCl 124, NaH$_2$PO4 1.2, KCl 2.5, $NaHCO_3$ 25, glucose 20, $CaCl_2$ 2, and $MgCl_2$ 1. They were then allowed to cool passively to room temperature. All slices were prepared by the same experimenter (HP), who followed the same procedure on each day.

## Recording methods

Whole-cell patch-clamp recordings were made between 1 to 10 hr after slice preparation using procedures described previously (*Pastoll et al., 2013*; *Pastoll et al., 2012a*; *Pastoll et al., 2012b*; *Sürmeli et al., 2015*). Recordings were made from slice perfused in standard ACSF maintained at 34–36°C. In these conditions, we observe spontaneous fast inhibitory and excitatory postsynaptic potentials, but do not find evidence for tonic GABAergic or glutamatergic currents. Patch electrodes were filled with the following intracellular solution (in mM): K gluconate 130; KCl 10, HEPES 10, $MgCl_2$ 2, EGTA 0.1, $Na_2ATP$ 2, $Na_2GTP$ 0.3 and NaPhosphocreatine 10. The open tip resistance was 4–5 MΩ, all seal resistances were >2 GΩ and series resistances were <30 MΩ. Recordings were made in current clamp mode using Multiclamp 700B amplifiers (Molecular Devices, Sunnyvale, CA, USA) connected to PCs via Instrutech ITC-18 interfaces (HEKA Elektronik, Lambrecht, Germany) and using Axograph X acquisition software (http://axographx.com/). Pipette capacitance and series resistances were compensated using the capacitance neutralization and bridge-balance amplifier controls. An experimentally measured liquid junction potential of 12.9 mV was not corrected for. Stellate cells were identified by their large sag response and the characteristic waveform of their action potential after hyperpolarization (see *Alonso and Klink, 1993*; *Gonzalez-Sulser et al., 2014*; *Nolan et al., 2007*; *Pastoll et al., 2012a*).

To maximize the number of cells recorded per animal, we adopted two strategies. First, to minimize the time required to obtain data from each recorded cell, we measured electrophysiological features using a series of three short protocols following initiation of stable whole-cell recordings. We used responses to sub-threshold current steps to estimate passive membrane properties (*Figure 2A*), a frequency modulated sinusoidal current waveform (ZAP waveform) to estimate impedance amplitude profiles (*Figure 2B*), and a linear current ramp to estimate the action potential threshold and firing properties (*Figure 2C*). From analysis of data obtained with these protocols, we obtained 12 quantitative measures that describe the sub- and supra-threshold integrative properties of each recorded cell (*Figure 2A–C*). Second, for the majority of mice, two experimenters made recordings in parallel from neurons in two sagittal brain sections from the same hemisphere. The median dorsal-ventral extent of the recorded SCs was 1644 μm (range 0–2464 μm). Each experimenter aimed to sample recording locations evenly across the dorsoventral extent of the MEC, and for most animals, each experimenter recorded sequentially from opposite extremes of the dorsoventral axis. Each experimenter varied the starting location for recording between animals. For several features, the direction of recording affected their measured dependence on dorsoventral location, but the overall dependence of these features on dorsoventral location was robust to this effect (*Supplementary file 9*).

## Measurement of electrophysiological features and neuronal location

Electrophysiological recordings were analyzed in Matlab (Mathworks) using a custom-written semi-automated pipeline. We defined each feature as follows (see also *Nolan et al., 2007*; *Pastoll et al., 2012a*): resting membrane potential was the mean of the membrane potential during the 1 s prior to injection of the current steps used to assess subthreshold properties; input resistance was the steady-state voltage response to the negative current steps divided by their amplitude; membrane time constant was the time constant of an exponential function fit to the initial phase of membrane potential responses to the negative current steps; the sag coefficient was the steady-state divided by the peak membrane potential response to the negative current steps; resonance frequency was the frequency at which the peak membrane potential impedance was found to occur; resonance magnitude was the ratio between the peak impedance and the impedance at a frequency of 1 Hz; action potential threshold was calculated from responses to positive current ramps as the membrane potential at which the first derivative of the membrane potential crossed 20 mv ms$^{-1}$ averaged across the first five spikes, or fewer if fewer spikes were triggered; rheobase was the ramp current at the point when the threshold was crossed on the first spike; spike maximum was the mean peak amplitude of the action potentials triggered by the positive current ramp; spike width was the duration of the action potentials measured at the voltage corresponding to the midpoint between the spike threshold and spike maximum; the AHP minimum was the negative peak membrane potential of the after hyperpolarization following the first action potential when a second action potential also occurred; and the F-I slope was the linear slope of the relationship between the spike rate and the injected ramp current over a 500 ms window.

The location of each recorded neuron was estimated as described previously (*Garden et al., 2008*; *Pastoll et al., 2012b*). Following each recording, a low magnification image was taken of the slice with the patch-clamp electrode at the recording location. The image was calibrated and then the distance measured from the dorsal border of the MEC along the border of layers 1 and 2 to the location of the recorded cell.

## Analysis of location-dependence, experience-dependence and inter-animal differences

Analyses of location-dependence and inter-animal differences used R 3.4.3 (R Core Team, Vienna, Austria) and R Studio 1.1.383 (R Studio Inc, Boston, MA).

To fit linear mixed effect models, we used the lme4 package (*Bates et al., 2014*). Features of interest were included as fixed effects and animal identity was included as a random effect. All reported analyses are for models with the minimal a priori random effect structure, in other words the random effect was specified with uncorrelated slope and intercept. We also evaluated models in which only the intercept, or correlated intercept and slope were specified as the random effect. Model assessment was performed using Akaike Information Criterion (AIC) scores. In general, models with either random slope and intercept, or correlated random slope and intercept, had lower AIC scores than random intercept only models, indicating a better fit to the data. We used the former set of models for all analyses of all properties for consistency and because a maximal effect structure may be preferable on theoretical grounds (*Barr et al., 2013*). We evaluated assumptions including linearity, normality, homoscedasticity and influential data points. For some features, we found modest deviations from these assumptions that could be remedied by handling non-linearity in the data using a copula transformation. Model fits were similar following transformation of the data. However, we focus here on analyses of the untransformed data because the physical interpretation of the resulting values for data points is clearer.

To evaluate the location-dependence of a given feature, p-values were calculated using a $\chi^2$ test comparing models to null models with no location information but identical random effect specification. To calculate marginal and conditional $R^2$ of mixed effect models, we used the MuMin package (*Bartoń, 2014*). To evaluate additional fixed effects, we used Type II Wald $\chi^2$ test tests provided by the car package (*Fox and Weisberg, 2018*). To compare mixed effect with equivalent linear models, we used a $\chi^2$ test to compare the calculated deviance for each model. For clarity, we have reported key statistics in the main text and provide full test statistics in the Supplemental Tables. In addition, the code from which the analyses can be fully reproduced is available at https://github.com/

MattNolanLab/Inter_Intra_Variation (*Nolan, 2020*; copy archived at https://github.com/elifescien-ces-publications/Inter_Intra_Variation).

To evaluate partial correlations between features, we used the function cor2pcor from the R package corpcor (*Schafer et al., 2017*). Principal components analysis used core R functions.

## Implementation of tests for modularity

To establish statistical tests to distinguish 'modular' from 'continuous' distributions given relatively few observations, we classified datasets as continuous or modular by modifying the gap statistic algorithm (*Tibshirani et al., 2001*). The gap statistic estimates the number of clusters ($k_{est}$) that best account for the data in any given dataset (*Figure 1—figure supplement 1A-C*). However, this estimate may be prone to false positives, particularly where the numbers of observations are low. We therefore introduced a thresholding mechanism for tuning the sensitivity of the algorithm so that the false-positive rate, which is the rate of misclassifying datasets drawn from continuous (uniform) distributions as 'modular', is low, constant across different numbers of cluster modes and insensitive to dataset size (*Figure 1—figure supplement 1D-G*). With this approach, we are able to estimate whether a dataset is best described as lacking modularity ($k_{est} = 1$), or having a given number of modes ($k_{est} > 1$). Below, we describe tests carried out to validate the approach.

To illustrate the sensitivity and specificity of the modified gap statistic algorithm, we applied it to simulated datasets drawn either from a uniform distribution ($k = 1$, $n = 40$) or from a bimodal distribution with separation between the modes of five standard deviations ($k = 2$, $n = 40$, sigma = 5) (*Figure 1—figure supplement 2A*). We set the thresholding mechanism so that $k_{est}$ for each distinct k (where $k_{est} \geq 2$) has a false-positive rate of 0.01. In line with this, testing for $2 \leq k_{est} \leq 8$ (the maximum k expected to occur in grid spacing in the MEC), across multiple (N = 1000) simulated datasets drawn from the uniform distribution, produced a low false-positive rate ($P(k_{est}) \geq 2 = {\sim}0.07$), whereas when the data were drawn from the bimodal distribution, the ability to detect modular organization ($p_{detect}$) was good ($P[k_{est}] \geq 2 = {\sim}0.8$) (*Figure 1—figure supplement 2B*). The performance of the statistic improved with larger separation between clusters and with greater numbers of data points per dataset (*Figure 1—figure supplement 2C*) and is relatively insensitive to the numbers of clusters (*Figure 1—figure supplement 2D*). The algorithm maintains high rates of $p_{detect}$ when modes have varying densities and when sigma between modes varies in a manner similar to grid spacing data (*Figure 1—figure supplement 3*).

## Analysis of extracellular recording data from other laboratories

Recently described algorithms (*Giocomo et al., 2014*; *Stensola et al., 2012*) address the problem of identifying modularity when data are sampled from multiple locations and data values vary as a function of location, as is the case for the mean spacing of grid fields for cells at different dorsoventral locations recorded in behaving animals using tetrodes (*Hafting et al., 2005*). They generate log-normalized discontinuity (which we refer to here as lnDS) or discreteness scores, which are the log of the ratio of discontinuity or discreteness scores for the data points of interest and for the sampling locations, with positive values interpreted as evidence for clustering (*Giocomo et al., 2014*; *Stensola et al., 2012*). However, in simulations of datasets generated from a uniform distribution with evenly spaced recording locations, we find that the lnDS is always greater than zero (*Figure 1—figure supplement 4A*). This is because evenly spaced locations result in a discontinuity score that approaches zero, and therefore the log ratio of the discontinuity of the data to this score will be positive. Thus, for evenly spaced data, the lnDS is guaranteed to produce false-positive results. When locations are instead sampled from a uniform distribution, approximately half of the simulated datasets have a log discontinuity ratio greater than 0 (*Figure 1—figure supplement 4A*), which in previous studies would be interpreted as evidence of modularity (*Giocomo et al., 2014*). Similar discrepancies arise for the discreteness measure (*Stensola et al., 2012*). To address these issues, we introduced a log discontinuity ratio threshold, so that the discontinuity method could be matched to produce a similar false-positive rate to the adapted gap statistic algorithm used in the example above. After including this modification, we found that for a given false-positive rate, the adapted gap statistic is more sensitive at detecting modularity in the simulated datasets (*Figure 4—figure supplement 1B*).

To establish whether the modified gap statistic detects clustering in experimental data, we applied it to previously published grid cell data recorded with tetrodes from awake behaving animals (*Stensola et al., 2012*). We find that the modified gap statistic identified clustered grid spacing for 6 of 7 animals previously identified as having grid modules and with $n \geq 20$. For these animals, the number of modules was similar (but not always identical) to the number of previously identified modules (*Figure 1—figure supplement 5*). By contrast, the modified gap statistic does not identify clustering in five of six sets of recording locations, confirming that the grid clustering is likely not a result of uneven sampling of locations (we could not test the seventh as location data were not available). The thresholded discontinuity score also detects clustering in the same five of the six tested sets of grid data. From the six grid datasets detected as clustered with the modified gap statistic, we estimated the separation between clusters by fitting the data with a mixture of Gaussians, with the number of modes set by the value of k obtained with the modified gap statistic. This analysis suggested that the largest spacing between contiguous modules in each mouse is always >5.6 standard deviations (mean = 20.5 ± 5.0 standard deviations). Thus, the modified gap statistic detects modularity within the grid system and indicates that previous descriptions of grid modularity are, in general, robust to the possibility of false positives associated with the discreteness and discontinuity methods.

## Acknowledgements

We thank Vanessa Stempel for comments on the manuscript, Tor Stensola and Edvard Moser for sharing published data, and Lukas Solanka and Lukas Fischer for help with building the large cage. This work was supported by grants to MN from the Wellcome Trust (200855/Z/16/Z) and the BBSRC (BB/L010496/1, BB/1022147/1 and BB/H020284/1).

## Additional information

### Funding

| Funder | Grant reference number | Author |
| --- | --- | --- |
| Biotechnology and Biological Sciences Research Council (BBSRC) | 200855/Z/16/Z | Matthew F Nolan |
| Biotechnology and Biological Sciences Research Council (BBSRC) | BB/1022147/1 | Matthew F Nolan |
| Biotechnology and Biological Sciences Research Council | BB/H020284/1 | Matthew F Nolan |
| Wellcome | 200855/Z/16/Z | Matthew F Nolan |

The funders had no role in study design, data collection and interpretation, or the decision to submit the work for publication.

### Author contributions

Hugh Pastoll, Conceptualization, Data curation, Software, Formal analysis, Validation, Investigation, Visualization, Methodology, Writing - original draft, Writing - review and editing; Derek L Garden, Formal Analysis, Investigation, Writing - reviewing and editing.; Ioannis Papastathopoulos, Formal analysis, Writing - review and editing; Gülşen Sürmeli, Resources, Methodology, Writing - review and editing; Matthew F Nolan, Conceptualization, Formal analysis, Supervision, Funding acquisition, Visualization, Writing - original draft, Project administration, Writing - review and editing

### Author ORCIDs

Derek L Garden https://orcid.org/0000-0003-3336-3791
Matthew F Nolan https://orcid.org/0000-0003-1062-6501

## Ethics

Animal experimentation: All experimental procedures were performed under a United Kingdom Home Office license (PC198F2A0) and with approval of the University of Edinburgh's animal welfare committee.

## Decision letter and Author response

Decision letter https://doi.org/10.7554/eLife.52258.sa1
Author response https://doi.org/10.7554/eLife.52258.sa2

# Additional files

## Supplementary files

• Supplementary file 1. Dependence of calbindin cell properties on dorsoventral position. Analyses are as described for *Table 1*. Data are from GFP-positive putative pyramidal neurons (n = 42, N = 3).

• Supplementary file 2. Dependence of SC properties on age. The distinguishing electrophysiological features of SCs and their dorsoventral organization were apparent at all ages, with some features depending significantly on age (left columns), consistent with the idea that SCs continue to mature beyond P18 (*Boehlen et al., 2010*; *Burton et al., 2008*). When we considered only animals between P33 and P44, we did not find any significant effect of age (right columns). Significance estimates for the effects of dorsoventral position (dvloc), age (age) and interactions between dorsoventral position and age (dvloc:age) were estimated using type II ANOVA and Wald $\chi^2$ test from fits to mixed models containing age and location as fixed effects and animal identity as random effects. Significance estimates were adjusted for multiple comparisons using the Benjamini and Hochberg method.

• Supplementary file 3. Dependence of SC properties on housing. Analyses suggesting that the membrane potential sag, resonance frequency, and spike half-width of SCs differ between mice housed in standard and large home cages. Significance estimates for the effects of dorsoventral position (dvloc), housing (housing) and interactions between dorsoventral position and housing (dvloc:housing) estimated using type II ANOVA and Wald $\chi^2$ test from fits to mixed models containing age and location as fixed effects and animal identity as random effects. Initial significance estimates (raw p) were adjusted for multiple comparisons (adjusted p) using the Benjamini and Hochberg method.

• Supplementary file 4. Inter-animal differences in electrophysiological features remain after accounting for housing. Results from comparison of a mixed effect model incorporating dorsoventral location and housing with an equivalent linear model. The significance estimate (p) is calculated using a $\chi^2$ test and adjusted for multiple comparisons (p_adj) using the Benjamini and Hochberg method.

• Supplementary file 5. Dependence of SC properties on hemisphere. We did not find significant effects of brain hemisphere on any features except for the relationship between dorsoventral location and sag. Significance estimates for the effects of dorsoventral position (dvloc), brain hemisphere (hemi) and interactions between dorsoventral position and hemisphere (dvloc:hemi) were estimated using type II ANOVA and Wald $\chi^2$ test from fits to mixed models containing age and location as fixed effects and animal identity as random effects. Initial significance estimates (raw p) were adjusted for multiple comparisons (adjusted p) using the Benjamini and Hochberg method.

• Supplementary file 6. Dependence of SC properties on mediolateral position. Mediolateral as well as dorsoventral position has been reported to determine the sub-threshold electrophysiological features of SCs (*Canto and Witter, 2012*). We found significant effects of mediolateral position on all measured electrophysiological features. However, the sizes of the effects of mediolateral position on subthreshold features (vm, ir, sag, tau, resf, resmag, and rheo) were much smaller than for dorsoventral position. By contrast, supra-threshold features (spkthr, spkmax, and ahp) were more greatly affected by mediolateral position, with more medial neurons having a higher spike threshold, and lower amplitudes of the spike peak and of after-hyperpolarization. Fixed effects are the intercept and slope coefficients for mixed models containing dorsoventral and mediolateral location as fixed effects and animal identity as random effects. Significance estimates for the effects of dorsoventral

position (dvloc), mediolateral position (ml) and interactions between dorsoventral position and mediolateral position (dvloc:ml) are estimated using type II ANOVA and Wald $\chi^2$ tests from the fits of the mixed models. Initial significance estimates (raw p) were adjusted for multiple comparisons (adjusted p) using the Benjamini and Hochberg method.

• Supplementary file 7. Dependence of SC properties on experimenter. We found that for many electrophysiological features, the identity of the experimenter affected the intercept, but not the slope, of their relationship with dorsoventral position. All features except for spike threshold nevertheless followed a dorsoventral organization after accounting for the experimenter. Significance estimates for the effects of dorsoventral position (dvloc), experimenter (exp) and interactions between dorsoventral position and experimenter (dvloc:exp) were estimated using type II ANOVA and Wald $\chi^2$ tests from fits to mixed models containing age and location as fixed effects and animal identity as random effects. Initial significance estimates (raw p) were adjusted for multiple comparisons (adjusted p) using the Benjamini and Hochberg method.

• Supplementary file 8. Dependence of SC properties on time since slice preparation. We anticipated that the interval between slice preparation and recording may influence the measured electrophysiological features. Consistent with our expectation, analyses of the data were consistent with changes to some electrophysiological features of SCs with time since slice preparation, but dorsoventral gradients could not be explained by these changes. Significance estimates for the effects of dorsoventral position (dvloc), time since slice preparation (rect) and interactions between dorsoventral position and experimenter (dvloc:rect) estimated using type II ANOVA and Wald $\chi^2$ tests from fits to mixed models containing age and location as fixed effects and animal identity as random effects. Initial significance estimates (raw p) were adjusted for multiple comparisons (adjusted p) using the Benjamini and Hochberg method.

• Supplementary file 9. Dependence of SC properties on direction in which sequential recordings are made. In anticipation of the effects of the time since slice preparation on the electrophysiological features of SCS, we varied the direction along the dorsoventral axis from which consecutive recordings were made between experimenters and experimental days (see 'Materials and methods'). Consistent with effects of time on electrophysiological features (see *Supplementary file 7* above), we found that the direction in which sequential recordings were made influenced the slope, but not the intercept of several electrophysiological features. Significance estimates for the effects of dorsoventral position (dvloc), direction in which sequential recordings were made (dir) and interactions between dorsoventral position and recording direction (dvloc:dir) estimated using type II ANOVA and Wald $\chi^2$ tests from fits to mixed models containing age and location as fixed effects and animal identity as random effects. Initial significance estimates (raw p) were adjusted for multiple comparisons (adjusted p) using the Benjamini and Hochberg method.

• Supplementary file 10. Inter-animal differences in extended models. Results from comparison of a mixed effect model incorporating dorsoventral location, housing, mediolateral position, experimenter identity and direction in which recordings were obtained with an equivalent linear model. Data are from animals between 32 and 45 days old. The significance estimate (p) is calculated using a $\chi^2$ test and adjusted for multiple comparisons (p_adj) using the Benjamini and Hochberg method.

• Supplementary file 11. Inter-animal differences in models fit to minimal datasets. Results from comparison of mixed effect models with dorsoventral location as a fixed effect and animal identity as a random effect using minimal datasets obtained by either HP (upper) or DG (lower). Data are from animals between 32 and 45 days old. Because of the smaller size of these datasets, the statistical power to detect inter-animal variation is reduced. Nevertheless, in these analyses, the conditional $R^2$ of the mixed model fit was again substantially higher than the marginal $R^2$, and most (9/12) features were better fit by a mixed model compared to a corresponding linear model in both datasets.

• Supplementary file 12. Electrophysiological features and the number of recorded neurons. Significance estimates for the effects of dorsoventral position (dvloc), number of recorded neurons (counts) and interactions between dorsoventral position and number of recorded neurons (dvloc:counts) estimated using type II ANOVA and Wald $\chi^2$ tests from fits to mixed models containing age and location as fixed effects and animal identity as random effects. Initial significance estimates (raw p) were adjusted for multiple comparisons (adjusted p) using the Benjamini and Hochberg method.

• Supplementary file 13. Inter-animal differences for experiments with >35 recorded neurons. Analyses of inter-animal differences focusing only on data from animals for which > 35 recordings were made (N = 11, n = 459). Comparison of marginal and conditional $R^2$ values continued to indicate substantial inter-animal variance, and fits obtained with mixed models remained significantly different to fits that did not account for animal identity ($p<4.4\times10^{-5}$). Analyses are as for *Supplementary file 1*, but are restricted to experiments in which > 35 neurons were recorded from.

• Supplementary file 14. Dependence of principal components on dorsoventral position and animal identity. Analyses are as described for *Table 1*, but were applied to principal components of the electrophysiological features of SCs.

• Supplementary file 15. Dependence of principal components of SC properties on housing. Analyses are as described for *Supplementary file 3*, but are applied to principal components of the electrophysiological features of SCs.

• Supplementary file 16. Dependence of principal components on animal identity in models that account for housing. Analyses are as for *Supplementary file 10*, but are applied to principal components of the electrophysiological features of SCs.

• Transparent reporting form

### Data availability

Processed data used for analyses and all associated code is available from the GitHub page for the project (https://github.com/MattNolanLab/Inter_Intra_Variation, copy archived at https://github.com/elifesciences-publications/Inter_Intra_Variation). Raw data has been made available from our institutional repository and can be found at https://doi.org/10.7488/ds/2765.

The following dataset was generated:

| Author(s) | Year | Dataset title | Dataset URL | Database and Identifier |
|---|---|---|---|---|
| Hugh P, Derek LG, Matthew FN | 2020 | Inter- and intra-animal variation in the integrative properties of stellate cells in the medial entorhinal cortex | https://doi.org/10.7488/ds/2765 | Edinburgh DataShare, 10.7488/ds/2765 |

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
