## [Decision Letter]

**Acceptance summary:**

In this study, the authors examined the electrophysiological properties of entorhinal cortical layer II stellate cells along the dorsal-ventral axis both within and across animals. They confirm, as previously reported, that a number of stellate cell biophysical features differ along this axis. Using sophisticated statistical methods, the authors then ask if these features follow a continuum pattern or a modular pattern along the dorsal-ventral axis. This has been an important question in the field since a discrete modular/clustered distribution of grid cell spacing was first described (Barry et al., 2007) and later work showed discrete modules (populations) of neurons with shared features of grid cell spacing, orientation, and elliptical shape (Stensola et al., 2012). The authors of the current paper systematically show that even with large numbers of neurons from single animals (N>35) they do not find evidence of modular/clustered patterns in the intrinsic properties of entorhinal stellate cells. The authors' findings add to the notion that the module cell organization of grid cells may reflect microcircuit activity. This is an important and rigorous paper that provides an important addition to the field and contributes to our understanding of the generation of spatial coding and circuits in the MEC.

**Decision letter after peer review:**

Thank you for submitting your article "Inter-and intra-animal variation of integrative properties of stellate cells in the medial entorhinal cortex" for consideration by *eLife*. Your article has been reviewed by three peer reviewers, and the evaluation has been overseen by a Reviewing Editor and Laura Colgin as the Senior Editor. The following individuals involved in review of your submission have agreed to reveal their identity: Michael E. Hasselmo (Reviewer #2); Andrea Burgalossi (Reviewer #3).

The reviewers have discussed the reviews with one another and the Reviewing Editor has drafted this decision to help you prepare a revised submission.

Essential revisions:

1) The authors have also studied the inter-animal variability among stellate cell excitability. They find that there is considerable inter animal variability, which is not unexpected given that there are a number of uncontrollable factors such as social interaction between animals as the authors allude to in their discussion. A number of studies have also reported that neuronal properties differ between animals at a single cell level in diverse regions (e.g. Villette et al., 2017). One possible issue for this variability, which the authors should address, is the fact that the slice quality, and therefore cell quality, might differ substantially between animals, even if this procedure is performed by the same experimenter. In particular, slight changes in cutting angles might impact factors such as the number of dendrites that are retained or cut off. Additional analyses on this front should be performed and considered in the discussion. Further, the synaptic connections onto the cells might also be variable between slices. I note that there are no glutamate or GABA receptor inhibitors included and thus baseline synaptic activity might be different between slices which might impact electrophysiological properties.

2) Given that the larger housing does have an effect on cell properties (Figure 4) it would be important to perform further analysis to determine if the number of objects that were in the cages within a specified area were the same in the larger housing versus standard cages. Additional objects in cages has been proposed to be an important factor in altering the properties of, for example, dentate gyrus adult new-born neurons.

3) The reviewers noted several discussion points that should be included in the Discussion section or relevant Results sections:

a) It is accepted that stellate cells are directly or indirectly involved in the generation of grid cell firing. However, grid cells are still a minority of all units recorded in MEC (and even in L2). By far, the most abundant functional cell type are non-grid spatial neurons (e.g. Diehl et al., 2017). Additional discussion on this point in relation to how their findings impact grid cell models should be considered.

b) One conclusion from the present work is that the intrinsic properties of stellate cells are not modularly organized – hence, the anatomical/cellular correlate of the functional grid cells modules remains to be elucidated. However, the L2 calbindin-positive neurons are indeed modularly organized, since they are anatomically clustered. This is an important organization principle of MEC L2 should at least be discussed (or commented upon) in the discussion.

c) Could the authors include a distribution of recordings along the medial-lateral axis?

d) Supplementary file 6: The physiological properties of neurons along the mediolateral dimension were also analyzed in Yoshida, Jochems and Hasselmo, 2013. This article should be cited and discussed. In addition, Figure 2, the apparent absence of dorsoventral differences in AHP differs from previous data showing a significant dorsoventral gradient of spike frequency adaptation and mAHP in the work by Yoshida et al., 2013. They should address this apparent difference in results.

---

## [Author Response]

Essential revisions:1) The authors have also studied the inter-animal variability among stellate cell excitability. They find that there is considerable inter animal variability, which is not unexpected given that there are a number of uncontrollable factors such as social interaction between animals as the authors allude to in their discussion. A number of studies have also reported that neuronal properties differ between animals at a single cell level in diverse regions (e.g. Villette et al., 2017).

We now mention results from the study by Villete et al. (paragraph two subsection “Functional consequences of within cell type inter-animal variability”). We note however that this study describes inter-animal differences in motor behaviour and activity of hippocampal neurons but does not address the neuronal properties that we investigate here. We’re not aware of previous studies that have systematically explored inter-animal variation in the intrinsic properties of mammalian neurons.

One possible issue for this variability, which the authors should address, is the fact that the slice quality, and therefore cell quality, might differ substantially between animals, even if this procedure is performed by the same experimenter. In particular, slight changes in cutting angles might impact factors such as the number of dendrites that are retained or cut off. Additional analyses on this front should be performed and considered in the discussion.

To address the possibility of variation in slice quality we now highlight in the results the standardisation of the slicing procedure and several observations that argue against there being substantial differences in slice quality (paragraph two subsection “Inter-animal differences remain after accounting for additional experimental parameters”).

We appreciate the possibility that variation in slice angle could in principle alter the number of proportion of dendrites that are cut off. This appears unlikely to be an issue for several reasons:

1) Our method for preparation of sagittal slices involves hemi-section along the midline of the brain and then gluing the cut surface of the brain to the cutting block. Because the midline of the brain serves is a reliable guide for orientation, variation is minimal. This is in contrast with preparation of horizontal slices for which there is no obvious landmark to use for orientation and is therefore harder to make reproducible.

2) The same experimenter (HP) prepared all of the slices.

3) After each patch-clamp experiment we took images of the slice. While the images were primarily to enable assessment of the dorsoventral location of the recorded neurons, they did not suggest any substantial variation in slice angle.

4) Stellate cells dendrites are oriented in all directions from the soma. Therefore we expect the number of dendrites to be independent of the slicing angle. In other ongoing work we have found this to be the case.

We have modified the Materials and methods section to highlight how are dissection procedure minimises potential issues related to variation in the slice angle (subsection “Slice preparation”).

Further, the synaptic connections onto the cells might also be variable between slices. I note that there are no glutamate or GABA receptor inhibitors included and thus baseline synaptic activity might be different between slices which might impact electrophysiological properties.

To address possible roles of glutamate or GABA receptor inputs we have carried out additional experiments.

To detect tonic and phasic GABA receptor mediated synaptic inputs we have recorded spontaneous synaptic activity and the holding current at a potential of -70 mV in conditions in which the Cl^–^ equilibrium potential is 0 mV (symmetrical intra- and extracellular Cl^-^concentration) (Author response image 1). This is different to our standard recording conditions but is designed to maximise the amplitude of any GABA-A receptor mediated effects. In these conditions we find that application of GABA-A receptor antagonists abolishes spontaneous inhibitory synaptic currents but causes only a very small tonic outward current (6.4 +/- 14.7 pA). Consistent with this, we did not find any detectable change in input resistance when we applied blockers of GABA-A receptors to stellate cells (control: 31.6 ± 3.6 MΩ, GABAzine: 31.9 ± 4.1 MΩ, p = 0.89, n = 6, N = 5). Similar experiments using blockers of AMPA and NMDA suggest there is very little tonic glutamatergic input to stellate cells in our standard slice conditions (change in holding current: 8.5 +/- 13.8 pA; control input resistance: 30.0 ± 5.8 MΩ, input resistance in NBQΧ and D-APV: 31.4 ± 5.6 MΩ, n = 2, N = 2).

These data argue against baseline synaptic activity explaining our results. We now address this in the Materials and methods.

**Author response image 1. respfig1:** Absence of a tonic GABA-A receptor mediated input to entorhinal cortex stellate cells. (**A–B**) Continuous voltage-clamp recording of spontaneous synaptic activity from a stellate cell in control conditions (**A**) and during perfusion of GABAzine to block GABA-A receptors (**B**). (**C–D**) Perfusion of GABAzine did not affect the holding current (**C–D**) or input resistance (**E**) but abolished fast spontaneous inhibitory currents (**F**).

2) Given that the larger housing does have an effect on cell properties (Figure 4) it would be important to perform further analysis to determine if the number of objects that were in the cages within a specified area were the same in the larger housing versus standard cages. Additional objects in cages has been proposed to be an important factor in altering the properties of, for example, dentate gyrus adult new-born neurons.

This is an interesting idea. There were 10 – 15 plastic objects and 8 cardboard rolls in the large cages. The standard cages contained one cardboard roll and did not contain plastic objects. The area ratio of the large cage to a standard cage is approximately 39:1 (standard cage dimensions of 20 x 37 cm; floor area of 0.074 m^2^ vs 2.88 m^2^ for the large cage). Therefore, while there were more objects in the large cage, their density was slightly lower (up to 1/0.125 m -2 vs 1/0.074 m^2^). We now add this information to the Materials and methods (subsection “Mouse strains”).

We appreciate the idea of investigating whether the number or density of objects influences intrinsic properties of stellate cells or other neurons. However, our experiments aimed to test whether the excitable properties of stellate cells are modifiable by an environment that would maximise activation of ventral grid cells. Because we did not attempt to systematically vary the number of objects in the large maze we are not able to analyse the effect of the number of objects separately from the size of the cage. This could be an important question for future studies. We note that because our data suggest that effect sizes when addressing this question are likely to be quite small, obtaining adequate statistical power will likely require a large scale study with many more animals in each experimental group than we use here. To address the reviewers’ point in the manuscript we have modified the Discussion to consider the role of the number of objects (paragraph three subsection “A conceptual framework for within cell type variability”).

3) The reviewers noted several discussion points that should be included in the Discussion section or relevant Results sections:a) It is accepted that stellate cells are directly or indirectly involved in the generation of grid cell firing. However, grid cells are still a minority of all units recorded in MEC (and even in L2). By far, the most abundant functional cell type are non-grid spatial neurons (e.g. Diehl et al., 2017). Additional discussion on this point in relation to how their findings impact grid cell models should be considered.

We have modified the Discussion to highlight this point (subsection “Implications of continuous dorsoventral organisation of stellate cell integrative properties for grid cell firing”).

b) One conclusion from the present work is that the intrinsic properties of stellate cells are not modularly organized – hence, the anatomical/cellular correlate of the functional grid cells modules remains to be elucidated. However, the L2 calbindin-positive neurons are indeed modularly organized, since they are anatomically clustered. This is an important organization principle of MEC L2 should at least be discussed (or commented upon) in the discussion.

We now mention this possibility in the Discussion (subsection “Implications of continuous dorsoventral organisation of stellate cell integrative properties for grid cell firing”).

c) Could the authors include a distribution of recordings along the medial-lateral axis?

Because our experiments used para-sagittal slices we only obtained two slices per hemisphere that contained the MEC. To show the distribution of data between these slices we now include a figure comparing properties from slices containing the more medial with the more lateral parts of the MEC (Figure 4—figure supplement 1, referred to on p 5, subsection “Inter-animal differences remain after accounting for additional experimental parameters”).

d) Supplementary file 6: The physiological properties of neurons along the mediolateral dimension were also analyzed in Yoshida, Jochems and Hasselmo, 2013. This article should be cited and discussed.

We apologise for overlooking the study by Yoshida and colleagues. We now cite this study (Results and Discussion section). We have added Discussion of the mediolateral organisation of SC properties (subsection “Implications of continuous dorsoventral organisation of stellate cell integrative properties for grid cell firing”).

In addition, Figure 2, the apparent absence of dorsoventral differences in AHP differs from previous data showing a significant dorsoventral gradient of spike frequency adaptation and mAHP in the work by Yoshida et al., 2013. They should address this apparent difference in results.

Differences between the results from Yoshida et al. and the results we present here likely result from different stimulus waveforms used to drive action potential firing. In Yoshida et al. spike frequency adaptation was measured in response to step currents that generated spike trains with frequency > 20 Hz and the action potential AHP was measured from a baseline membrane potential of -60 mV after triggering a single spike with a 1ms current step. In contrast, we measured action potentials activated during injection of a positive current ramp. Because we were using a current ramp we were unable to quantify adaptation. Moreover, the maximum spike frequency reached during the ramp was typically < 10 Hz (e.g. Figure 1C). Across this lower frequency range stellate cells show very little spike frequency adaptation (cf. Nolan et al., 2007, Pastoll et al., 2012). The dorsoventral gradient in the duration of AHPs reported by Yoshida et al. following single action potentials is consistent with our previous results in conditions similar to those used here (cf. Pastoll et al., 2012). We’re not sure why we don’t see the dorsoventral gradient in AHP amplitude reported by Yoshida et al., but this may reflect different driving forces for currents generating AHPs at the different baseline membrane potentials in each experiment (-60 mV in Yoshida et al. and approximately -40 mV here). Because the mechanisms by which SCs generate their AHP is not well understood it is difficult to speculate further. Therefore, because the two studies do not measure the AHP in the same way, and given that differences are likely to be technical and not related to the main hypotheses of the experiments in the study, we have not added any further discussion of the differences to the manuscript.